# Maternal Diet, Nutritional Status, and Birth-Related Factors Influencing Offspring’s Bone Mineral Density: A Narrative Review of Observational, Cohort, and Randomized Controlled Trials

**DOI:** 10.3390/nu13072302

**Published:** 2021-07-04

**Authors:** Daria Masztalerz-Kozubek, Monika A. Zielinska-Pukos, Jadwiga Hamulka

**Affiliations:** Department of Human Nutrition, Institute of Human Nutrition Sciences, Warsaw University of Life Sciences (SGGW-WULS), 02-787 Warsaw, Poland; daria_masztalerz_kozubek@sggw.edu.pl (D.M.-K.); monika_zielinska_pukos@sggw.edu.pl (M.A.Z.-P.)

**Keywords:** bone mineral density, bone health, bone development, osteoporosis, prenatal exposure, prenatal nutrition

## Abstract

There is growing evidence that bone health may be programmed in the first years of life. Factors during the prenatal period, especially maternal nutrition, may have an influence on offspring’s skeletal development and thus the risk of osteoporosis in further life, which is an increasing societal, health and economic burden. However, it is still inconclusive which early life factors are the most important and to what extent they may affect bone health. We searched through three databases (PubMed, Google Scholar, Cochrane Library) and after eligibility criteria were met, the results of 49 articles were analyzed. This narrative review is an overall summary of up-to-date studies on maternal diet, nutritional status, and birth-related factors that may affect offspring bone development, particularly bone mineral density (BMD). Maternal vitamin D status and diet in pregnancy, anthropometry and birth weight seem to influence BMD, however other factors such as subsequent growth may mediate these associations. Due to the ambiguity of the results in the analyzed studies, future, well-designed studies are needed to address the limitations of the present study.

## 1. Introduction

The first years of life are crucial for proper child development. For the first time, this theory was proposed by Baker and his hypothesis gave the grounds for the following concepts of metabolic programming, such as the first 1000 days of life or the Developmental Origins of the Health and Disease hypothesis (DOHaD) [1,2,3,4]. According to them, factors during periconceptual, fetal and early infant phases play an important role in a child’s growth and development. By consequence, if their influence is adverse, they may increase the risk of poorer neurocognitive development, metabolic disorders and related conditions, such as obesity, diabetes or cardiovascular diseases later in life [1,2,3,4,5]. Besides, it is also a very crucial period in bone development, which may influence bone health, including further risk of osteoporosis [4,6,7].

The human skeleton starts to develop in the early prenatal phase, when cells differentiate into chondrocytes and osteoblasts [8]. During the third trimester the majority of a fetus’ bone is gained, so this time seems to be very important in the terms of skeletal development [9]. The skeleton grows intensively in the utero environment and the rate of growth is still high immediately after birth, then slows and increases again in infancy. This continuous development lasts until about age 20 when peak bone mass is reached, but this time is dependent on skeletal site and sex, among others [8,9].

Genetic and epigenetic processes, as well as accessibility of nutrients, seems to play an important role in bone development and affect peak bone mass, whose lower levels are a crucial risk factor for osteoporosis [7,10,11]. Among dietary factors, vitamin D and calcium are the most often mentioned [11,12,13,14,15,16,17,18], however, the role of macronutrients [19,20,21] and other nutrients, such as vitamin A [22,23], folate [19,24] or magnesium [19,21] have also been raised in studies regarding bone health.

In addition, peak bone mass may be regulated by two components, skeletal envelope size and bone mineral density [25]. Bone mineral density (BMD) increases during childhood and adolescence and, according to the study of Foley et al. [26], BMD levels at age 8 may determine those at age 16. This is important from a long-term perspective, as it suggests that risk factors for osteoporosis may be identified in adolescence, when it is still possible to take actions aimed at improving skeletal health.

In the available studies several factors, both maternal (especially nutrition in pregnancy, particularly vitamin D status, maternal anthropometry), and birth-related (i.a. birth parameters) were analyzed in relation to offspring bone health. Also during the postnatal period, factors such as children nutrition (especially sufficient calcium intake and vitamin D status) or physical activity have a great contribution to offspring bone health and are considered as a part of primary prevention of osteoporosis [27]. Nevertheless, in the presented review we focus on the selected prenatal and birth-related factors.

The most frequently assessed parameters were: BMD, areal BMD (aBMD), bone mineral apparent density (BMAD), volumetric BMD (vBMD), and speed of sound (SOS). BMD is usually defined as bone mineral content (BMC)/bone surface area ratio [28]. However, it remains inconclusive whether and to what extent those factors may influence bone mineral density in children and if those changes are still apparent later in life. In addition, a recent systematic review [29] has raised a similar subject, but the authors included studies assessing BMD in young adults aged 16–30 years, so less is known about those associations in younger population. Moreover, the results of foregoing studies have been less consistent.

Taking into consideration the abovementioned background, the aim of this review was to identify and synthesize the best available evidence from observational, cohort, and randomized controlled trials on the association between maternal factors, especially nutritional and anthropometric, as well as birth-related factors influencing offspring bone health in the childhood up to young adulthood, particularly BMD—the main parameter analyzed in this review.

## 2. Methods

### 2.1. Search Strategy

In this narrative review a comprehensive literature search was performed until the 31 August 2020, with no starting date specified, using three databases, PubMed, Google Scholar and Cochrane Library, with no restrictions on type of studies, publication date, or country. Search terms were defined by two senior researchers (J.H., M.A.Z.-P.) and included the following keywords in the title and/or abstract: maternal, pregnancy, vitamin D, supplementation, status, diet, offspring, children, bone mass, BMI (body mass index), smoking, in various combinations. Due to the large number of results obtained in Google Scholar, in this database only the first few (10–15) pages were screened.

The search and eligibility process is presented in detail in Table 1 and Figure 1. After first screening by titles and further by titles and abstracts and removing duplicates, papers were screened in the full-text versions. Then, the exclusion criteria were applied (Table 1).

### 2.2. Analysis of Included Studies

In the assessment of the results we considered fully adjusted, multivariate models of conducted analyses. When such analysis had not been conducted, we focused on the available results of univariate models, correlation coefficients, or in intervention studies, comparisons between groups. Moreover, in the presented tables, if a particular factor/nutrient was mentioned in the column with the assessed factors but not in the column with the outcome, it means that this factor was not significant in the considered analysis.

## 3. Findings and Discussion

In total, 1867 studies were identified through the literature search up to 31 August 2020. Among these, 1044 were excluded in the first screening based on the title and 508 duplicates were removed. In the further eligibility process (Figure 1), we included 49 articles in the final review. We did not apply any criteria for the offspring’s age, thus the results of the analyzed studies include a wide age range, from neonates to young adults.

### 3.1. Maternal Factors

#### 3.1.1. Dietary Intake

The results of studies concerning maternal dietary intakes are presented in Table 2. In total, eight studies aimed to identify possible associations between maternal macronutrients, vitamins, mineral intake, as well as specified dietary patterns in pregnancy and offspring BMD. Furthermore, Appendix A provides detailed information about maternal dietary intakes reported in those studies, as well as methodological details about dietary assessment methods. In all of the assessed studies, the analyses focused on the assessment of the associations between quantitative data, expressed as daily intakes, quintiles [19] or tertiles [20,21]. In two studies [20,21], the authors converted dietary variables to nutrient/food density by dividing estimated daily nutrient/food intake by estimated total daily energy intake. Data about inadequate or excessive intake were not reported.

##### Macronutrients

Four studies investigated associations between BMD and dietary intake of components such as macronutrients [19,20,21,30]. Maternal energy from carbohydrate (quintiles) was negatively associated with offspring BMD [19]. In one study energy from protein (quintile) was positively related to BMD in offspring [19]. However, in two other studies, no association between protein intake was found [20,30]. Results regarding fat intake, expressed as quantiles of energy from fat [19] or fat density (divided estimated nutrient intake by estimated total daily energy intake) [20,21] were ambiguous: one study found a negative association between fat density and offspring lumbar spine (LS) and femoral neck (FN) BMD (with no changes in whole body (WB) BMD) [21], one study found a positive association [20], and one study reported no association [19].

##### Minerals

A possible link between children’s BMD and maternal dietary intake of minerals was assessed in six studies [19,20,21,30,31,32]. No association was found between offspring BMD and maternal potassium nor zinc intake in pregnancy [20,30]. In some studies, offspring BMD (in at least one site) was associated with higher maternal intakes of magnesium [21], phosphorus [19,20], calcium [19,21,31], or folate [32]. However, it remains inconclusive whether intake of those nutrients is associated with children’s bone outcomes, as the above results have not been confirmed in other studies [19,20,30].

##### Group Products

Associations between BMD and specific dietary pattern [33] or intake of group products like green vegetables [31], milk and milk products [24], or milk density (divided estimated daily food intake by estimated total daily energy intake) [21] were assessed in four studies. Interestingly, higher maternal green vegetable intake in early pregnancy was related to lower neonatal spine BMD [31]. Maternal milk and milk products intake as well as milk density in late pregnancy was positively associated with BMD in offspring [21,24]. In addition, Cole et al. [33] demonstrated that a high prudent diet score in late pregnancy was associated with higher offspring WB BMD.

##### Intervention Studies

Table 3 summarizes the results of dietary interventions during pregnancy. Three studies involved vitamin D supplementation [11,12,13]. Two studies did not find any associations between maternal vitamin D supplementation during pregnancy and neonatal BMD [11,12]. Contrary to those studies, the study conducted by Brustad et al. [13] demonstrated higher head BMD, with no difference in total body less head (TBLH) BMD or total BMD in the group of children (combined analysis at 3 and 6 years old) whose mothers received higher vitamin D_3_ doses.

In turn, in studies where intervention included supplementation of vitamin D and calcium, the results were conflicting [15,16]. In a study from India, WB BMD was higher in offspring whose mothers were in the placebo group, compared to groups that received high doses of vitamin D [15]. In another study no association was found.

Studies on the effect of supplementation with calcium [14] and n-3 long chain polyunsaturated fatty acids (LC PUFAs) [34] showed no association with offspring vBMD and TBLH BMD, respectively.

#### 3.1.2. Nutritional Status

##### Vitamin D Status

Maternal vitamin D status during pregnancy was assessed in eleven studies (Table 4), however the results are inconclusive. One study reported a negative association with offspring total BMD (with no difference in lumbar BMD) [17]. Six studies found no association between maternal vitamin D status and bone parameters from the neonatal period even to the age of 26 years [23,35,36,37,38,39]. In the four remaining studies, positive associations were found between maternal vitamin D status and offspring BMD [18,40], with sex-specific association in one study [41], as well as neonatal bone SOS [42].

##### Other Nutrients

Table 5 presents the results of studies whose aim was to assess maternal nutritional status in categories other than vitamin D. In one study, maternal vitamin B_12_ and homocysteine concentrations in pregnancy were assessed [19]. Vitamin B_12_ concentrations were positively associated with offspring BMD, whereas in relation to homocysteine no association was found [19]. Two studies investigated maternal folate concentrations but the results varied [19,24]. In one study no association was found [19], whereas in another study a positive association with offspring BMD was noticed [24]. In a study by Javaid et al. [18], calcium concentrations in cord blood were positively related to offspring LS aBMD but not WB aBMD. Two studies assessed maternal vitamin A status in relation to offspring BMD [22,23]. When samples were taken at around weeks 33/34 of gestation, no association was found in both studies [22,23]. However, retinol concentrations in samples taken in week 17 and 37 of gestation showed a positive association with WB, LS BMD and LS, total hip BMD, respectively [23]. Harvey et al. [43] investigated maternal LC PUFAs status. Late pregnancy concentrations of n-3 LC PUFAs, especially docosapentaenoic acid (DPA) and eicosapentaenoic acid (EPA), were positively associated with offspring aBMD/vBMD at several sites, whereas arachidonic acid (AA) was inversely related to WBMH aBMD, which suggests the significant role of n-6 to n-3 balance [43].


*Potential mechanisms of associations between maternal dietary intake, nutritional status and offspring bone outcomes*


There are several potential mechanisms that could explain associations between maternal dietary intake of the analyzed nutrients and offspring bone health. Higher protein intake may increase bone mineral accrual and insulin-like growth factor 1 (IGF-1) concentrations. IGF-1 is an osteotrophic factor, whose levels have been related to a higher BMD [19,44]. Saturated fat intake has been inversely related to bone density in adults [45], besides in animal studies a high intake of fat decreased intestinal calcium absorption and thus could have a negative impact on offspring bones [21]. Magnesium deficiency may have both a direct and indirect effect on bone health, through an influence on bone cells (reducing osteoblast and enhancing osteoclast activity) and changes in calcium homeostasis regulators (parathyroid hormone and 1,25(OH)_2_D, leading to hypocalcemia), as shown in in vitro and in vivo studies [46].

In all of the included studies on maternal vitamin D status, authors assessed 25(OH)D concentrations [17,18,23,35,36,37,38,39,40,41,42]. In addition, in one study, 1,25(OH)_2_D concentrations were also assessed and its levels were similar across different levels of 25(OH)D [23]. According to the guidelines, recommended marker for assessing vitamin D status is 25(OH)D, which is the most abundant metabolite of vitamin D [47,48]. Possible mechanisms given in the analyzed studies suggest an effect of vitamin D on fetal skeletal development and mineralization, which may be impaired because of this relevant vitamin deficiency [23,40]. Moreover vitamin D is essential for proper placental calcium transport, so its insufficient levels may affect the trajectory of bone mineral accrual in intrauterine environments [18]. However, since some of the studies did not show an association between maternal vitamin D status and children’s bone outcomes [23,35,36,37,38,39], there may be compensatory mechanisms that enable proper skeletal growth in the children of mothers with vitamin D deficiency during pregnancy [23,38].

Maternal LC PUFAs status/intake in pregnancy seems to have a positive influence on children’s bone outcomes, but the potential mechanism is uncertain. Essential n-6 and n-3 are substrates for the production of eicosanoids and thus may have a variety of effects in human bone cell culture, like inhibited osteoclast formation and enhanced osteoblast formation, which is optimal, as well as osteoblast apoptosis, which is adverse [49]. Vitamin A plays the role of modulator in epigenetic processes, moreover retinol is an important nutrient in embryogenesis, when the axial skeleton is shaped [23,50]. Carotenoids may be beneficial for bone health, as some of them are provitamin A precursors, they have antioxidant functions and reduce bone resorption [50]. However, some studies have observed that high intake of supplements or fortified foods with preformed vitamin A (retinol and retinyl esters) may be related to higher bone loss [50]. Folate and vitamin B_12_ may have an indirect impact on bone health through epigenetic changes, as they are important methyl donors for deoxyribonucleic acid (DNA) methylation, but also a direct effect related to osteoblast function or homocysteine metabolism is possible [19,51].

#### 3.1.3. Maternal Anthropometry

Regarding maternal anthropometry, the included studies are presented in Table 6 and Table 7. Maternal height was related to offspring bone parameters, but the results of five analyzed studies are inconclusive [24,31,52,53,54]. In one study, this maternal factor was inversely associated with spine BMAD, but no association was observed in relation to TB and spine BMD [53]. Three studies suggested no effects regarding offspring BMD [24,52,54]. Godfrey et al. [31], in turn, showed a positive association between maternal stature and neonatal spine BMD.

In one study, higher maternal early pregnancy fat stores, assessed by triceps skinfold thickness, were associated with higher neonatal WB BMD [31].

Maternal BMI in early pregnancy was analyzed in two studies. In one study, no association was found [31], whereas in another study a positive association was noticed but only in relation to one of the measured sites (LS BMD) [53]. The influence of pregnancy weight gain, which was investigated in three studies [53,55,56], was also ambiguous. In a study by Xu et al. [56], pregnancy weight gain was positively related to offspring BMD. In another study, a positive influence was observed only in relation to one measured site (TB BMD) [53]. Monjardino et al. [55] found that maternal gestational weight gain (GWG) was associated with higher aBMD in children, but only if the mothers were under/normal weight at the beginning of pregnancy.

Four studies assessed the possible link between maternal prepregnancy weight and offspring BMD (Table 7). In three of them, authors found no association [57,58,59], whereas Rudang et al. [54] found a positive association.


*Potential Mechanisms of Associations between Maternal Anthropometry and Offspring Bone Outcomes*


Maternal height may have an influence on children’s BMD through genetics. Moreover, larger pelvic diameter in mothers with a higher stature may be a factor that influences fetal growth because of a greater capacity to supply the fetus with nutrients [52]. Both triceps skinfold thickness and BMI are indicators of maternal fat stores, which may reflect overall maternal nutritional status and the availability of nutrients in the uterus, which contributes to fetal growth [31,59]. Nonetheless, it needs to be considered that in the Macdonald-Wallis et al. [59] study, the results between maternal and paternal BMI and bone outcomes were similar, so this effect might arise from genetics and the postnatal environment, rather than the environment in the uterus. Furthermore, in a study where DXA assessments were made several times over six years, maternal prepregnancy weight was not related to offspring BMD at each individual assessment [57]. This may indicate that maternal prepregnancy weight does not affect BMD in offspring in the first six years.

#### 3.1.4. Maternal Demographic and Socioeconomic Factors

Other maternal factors, such as age [54], education [60], socioeconomic status (SES) [61] and parity [24,52,54,62], were also analyzed regarding offspring BMD (Table 8).

Advancing maternal age was associated with lower spine aBMD in sons [54]. No influence of maternal education levels [60] nor parity [24,52,54,62] on childhood bone outcomes has been confirmed. One study investigated maternal SES and this factor was associated with higher BMD at all measured sites in daughters [61]. Identical results were obtained in sons with the exception of LS BMD (no effect) [61].


*Potential Mechanisms of Associations between Maternal Demographic, Socioeconomic Factors and Offspring Bone Outcomes*


A possible reason for the association between maternal age and offspring aBMD is epigenetic causes, related to DNA methylation and histone modification in utero [54], whereas maternal SES may be a factor that influences nutritional behaviors as well as anthropometric measures and thus affects bone growth [61].

#### 3.1.5. Maternal Smoking in Pregnancy

Eight studies analyzed maternal smoking in pregnancy [31,52,54,63,64,65,66,67] (Table 9). However, its influence was confirmed only in one study, where the authors have observed lower neonatal WB BMD [31]. Two studies suggested a positive association between maternal smoking and BMD, but those associations were no longer significant after adjusting several factors, such as current weight [66,67].


*Potential Mechanisms of Associations between Maternal Smoking in Pregnancy and Offspring Bone Outcomes*


Smoking in pregnancy has extremely harmful effects on a fetus, even if the negative effect on BMD was not observed in several studies [52,54,64,65]. Moreover, maternal smoking has been related to adverse effects on bone structure, so the negative influence of smoking in utero may be independent of BMD [63]. Adverse effects of maternal smoking are mostly caused by impaired placental functions and calcium transport, as well as the toxic effect of cadmium [31,52,68,69]. It is worth mentioning that according to some authors maternal smoking affects birth weight or weight gain later in life, and thus has an adverse indirect effect on bone parameters in offspring [65,66]. Moreover, a similar assumption was made in a recent systematic review by Jensen et al. [29].

### 3.2. Birth-Related Factors

#### 3.2.1. Birth Anthropometry

Birth anthropometry might be another factor that contributes to bone outcomes and was analyzed in fifteen studies, presented in Table 10 [10,23,24,31,42,53,54,56,62,63,64,65,70,71,72].

The ponderal index was analyzed in one study and was positively related to WB, but not spine BMD [31].

Four studies assessed birth length [24,56,62,70]. In two studies it was not related to offspring BMD [24,56], in another study a positive association was observed with neonatal bone SOS [62] and in the final study birth length was positively related to BMD at only one site [70].

Fourteen studies investigated birth weight [10,23,31,42,53,54,56,62,63,64,65,70,71,72]. In one study, neonatal bone SOS was lower in infants with a lower birth weight in comparison to those with a higher one [42]. In addition, in the study by Heppe et al. [10], BMD was also lower in children with a lower birth weight, but authors found no association between weight to gestational age and BMD. Six studies found no association [23,54,63,64,70,72]. In five studies, birth weight was positively associated with offspring bone parameters (in one study only at one site [53]) [31,56,62,65]. Moreover, Akcakus et al. [71] found that children that were born large for gestational age (LGA) had higher BMD in comparison to those that were born small for gestational age (SGA).


*Potential Mechanisms of Associations between Birth Anthropometry and Offspring Bone Outcomes*


A possible mechanism of the association between birth anthropometry and bone outcomes may be the result of hormonal changes. Fetal growth restriction probably has a negative effect on the growth hormone/insulin-like growth factor 1 (GH/IGF-1) axis, which has been deemed an important determinant of bone mass acquisition [73]. Higher IGF-1 levels are related to higher bone mass and this may be a reason of lower bone mass in children who were born with a lower birth weight [74]. Moreover, neonates born with a lower birth weight had higher serum cortisol levels [75]. This hormone has been negatively associated with bone mass and its levels may be a determinant of prospectively determined bone loss [10,76]. However, the influence of birth anthropometry, especially birth weight, might be of less importance than subsequent growth, as after adjustment for this factor, associations were no longer significant [63,64].

#### 3.2.2. Gestational Age

The influence of gestational age was investigated in seven studies, presented in Appendix A. In the study conducted by Bas et al. [77], children born preterm had lower BMD in comparison to those born at term. In two other studies, no association was found [10,24]. In the remaining four studies, gestational age was positively related to offspring bone parameters [31,42,60,62].

### 3.3. Ambiguity in Analyzed Studies

In total, the results of 49 studies were summarized in the presented review (Appendix A). Maternal vitamin D status and birth anthropometry were assessed in the most studies, 11 [17,18,23,35,36,37,38,39,40,41,42] and 15 [10,23,24,31,42,53,54,56,62,63,64,65,70,71,72], respectively. However, factors that seem to be the most relevant in terms of BMD are dietary intake, other than vitamin D nutrients status, and gestational age, as the plurality of the studies showed positive associations. Despite our major efforts, this review does not cover the entirety and complexity of research in this field and their conflicting results. Overall, this review demonstrates that the first years of life are important for proper bone development and that the analyzed early life factors may influence a child’s bone mineral density to some extent.

Inconclusive results in the studies related to maternal vitamin D status may arise from the fact that both bone measurements and maternal vitamin D levels were performed in different age ranges or week of gestation, respectively. It might be misleading, because 25(OH)D may have a different effect on bone parameters depending on the moment of pregnancy. On the one hand, the third trimester seems to be the most important in terms of bone development [9]. Essential nutrients like calcium and vitamin D are supplied to the fetus intensely in this period [6,12]. On the other hand, the results of Hyde et al. [30] suggest that maternal 25(OH)D levels in early pregnancy may also be very important. Nonetheless, the second trimester is also crucial, as the long bones accelerate their growth during this time [76].

The fact that that maternal vitamin D supplementation in pregnancy did not affect offspring BMD in some of the included studies may be explained in several ways. First of all, studies with null effect on BMD were performed with neonatal patients [11,12,16]. In a study that assessed BMD in older children, aged 3 and 6 years old, a higher head BMD was observed in the group of children whose mothers received 2800 IU of vitamin D [13]. As mentioned in the study by Diogenes et al. [16], infant BMD may decrease in the first months of life, which is a physiological effect, therefore the influence of maternal vitamin D supplementation in pregnancy might not be significant in the youngest children. In addition, this dependence may be supported by a recent systematic review, which reported that maternal 25(OH)D status may increase BMD in young adult offspring [29]. Another explanation is that the groups of mothers in the studies were not equal in terms of vitamin D status at the moment of enrollment to the trials. In addition, the doses of vitamin D and periods of interventions were different [11,12,13,15,16].

Inconsistency in the results of studies regarding maternal dietary intakes may arise from different methodologies (e.g., various FFQ) and the reference period of food intake assessment in the analyzed studies [19,20,21,24,30,31,32,33]. Admittedly, all of the included studies used FFQ, however questionnaires were adjusted to examined populations, thus there might be differences in obtained data, depending on the type of applied FFQ. Nonetheless, FFQs that were applied were in majority validated (Appendix A). In some studies [19,20,21,24,31,32,33] semiquantitative FFQs were used, thus authors could estimate quantities of foods eaten and/or nutrients intakes [78]. FFQ is a commonly used method to estimate selected food items usually eaten, which is characterized by the low cost of processing, respondent burden, and requires little time [78,79]. Moreover, FFQ is suitable for studies on large populations [78]. Nevertheless, it has limitations, including the possibility of inaccurate quantification of food intake, and requires memory of food patterns in the past. Taken together, results regarding maternal food intake during pregnancy should be interpreted cautiously.

It also remains inconclusive through which mechanisms maternal diet in pregnancy may play a role in offspring bone development. On the one hand, it may be due to individual nutrients, on the other hand, the effect of an overall healthier diet. Moreover, it has been suggested that maternal diet in pregnancy is more likely to have an impact on bone health by long-term metabolic programming than specific mechanisms in utero, as a very small amount of bone is laid down during pregnancy [20,21]. In addition, it needs to be considered that in the analyzed studies only quantitative data were assessed (Appendix A). Both inadequate and excessive nutrients intake may have adverse effects on bone health [17,50], nonetheless included studies aimed at the assessment of the association between dietary intake, not sufficient/insufficient intakes.

The influence of birth anthropometry on bone outcomes seems to be temporal, as changes were observed more often in neonates and young children than in older offspring [53,64,70,71]. Moreover, after adjusting for weight gain, differences were no longer significant [63,64]. For example, in the study by Ay et al. [53], catch-up weight in the first six weeks decreased the possibility of low BMD. So, even if a low birth weight may reflect an adverse intrauterine environment, and thus limited bone development in utero, this effect might be observed only in early childhood and may not be independent of weight gain later in life [63]. Furthermore, the results of systematic reviews suggest that birth weight affects adult BMD to a much lesser extent than BMC [80,81]. All things considered, it seems that weight and height gain later in life, as well as environmental factors, may be more important in terms of bone development, rather than birth anthropometry. In addition, it might not be possible to estimate the effects of birth weight, growth, and current weight independently of each other, because they are related [70]. However, there is still considerable ambiguity with regard to the influence of later weight and height, as Foley et al. [26] found that bone mass tracks largely independently of linear growth.

In addition, Choi et al. [82] showed an association between parental and offspring BMD, suggesting that peak bone mass acquisition may be influenced by genetic factors rather than environmental. Therefore, poor bone health may cause negative implications for further generations, as children of adults with lower BMD may also be at risk of lower BMD. This review has raised the need for further observations, especially longitudinal studies to assess the trajectory of changes in BMD levels during childhood and adolescence in regard to the analyzed factors.

### 3.4. Future Directions

The results of the included studies have several possible implications that should be taken into consideration. Because BMD levels in childhood may be related to those in adolescence [26], considerable attention should be paid to improving BMD levels in the first years of life. Especially considering that according to Jones et al. [70], bone growth trajectory is determined in this early life period. Moreover, BMD is one of the factors that contributes to peak bone mass, which has been suggested to be a predictor of osteoporosis risk in further life [7,25]. Taken together, the early life period may have a long-term influence on bone health. Because some of the assessed factors may be modifiable (i.a. nutrition or smoking during pregnancy), education programs directed at pregnant women would be beneficial, as those factors may also influence other aspects of a child’s development [2,83,84]. Hence, further attention should be paid to prevention strategies. These should focus on the improvement of nutritional status in women in the preconceptional and prenatal periods as well as on proper health care during pregnancy.

Future research is therefore required, especially well-designed studies. The obtained results of further high-quality studies would enable us to create recommendations aimed at improving bone health from the very first years of life, and in turn, to lower the risk of osteoporosis in the elderly.

#### Strengths and Limitations

The main strength of the presented review is that we analyzed many factors that have been suggested in previous studies to influence BMD. Moreover, we included studies that assessed not only BMD but also related parameters, such as aBMD, vBMD, and SOS, thus it was possible to obtain a broader picture of those associations. The fact that the presented study did not include criteria regarding age or method of bone assessment may be considered both a strength and a limitation. Nonetheless, our work clearly has some limitations, which need to be considered. First, our review does not meet the criteria for a full systematic review, which is why, despite our major effort in the search process, some studies might not have been included. Second, BMD may be underestimated in children because of their short stature and should be corrected for bone size (BMAD) [53]. Thus, BMAD may be more appropriate parameter than aBMD, which is derived from the bone area and may limit the use of DXA for example in children with abnormal growth patterns [85,86]. Nonetheless, in the study by Kalkwarf et al. [87], the authors demonstrated the practicability of measuring aBMD in children aged 1–3 years old, and they provided values for this parameter that can be used in the evaluation of bone deficits. Anyway, DXA measurements do not allow us to assess “true” density, as it would require the consideration of tridimensional bone depth. BMD value is in fact the BMC/bone area (BA) index and in DXA it is possible to obtain only a bidimensional BA [88]. Nevertheless, as BMD demonstrated a low variability during childhood, children with a low BMD are more likely to also have a low BMD later in life [86]. Taking into consideration the above, as well as that BMD may be related to fracture risk in healthy children and contributes to peak bone strength, BMD seems to be an important and evolving field in the children bone health assessment [8,86]. More interestingly, the results in one study have also suggested that TB BMD may be mostly associated with prenatal factors, whereas LS BMD with postnatal factors [53], and in our review we included various parameters. It should also be emphasized that some factors may influence bone development via other parameters than bone density. For example, the mechanism of the association between birth anthropometry may be through bone size rather than density, as birth anthropometry made significant contributions to BMD but not BMAD after adjustment for subsequent growth [70].

## 4. Conclusions

This review revealed that maternal and birth-related factors may influence children’s bone mineral density. Nonetheless, our paper has highlighted the inconsistencies in the foregoing studies. The findings of this review suggest that factors during prenatal period, such as maternal nutritional status, which may be a reflection of overall healthier diet, as well as factors related to gestation or birth, like gestational age and birth parameters, through direct and indirect mechanisms may contribute to offspring’s bone health. This paper has underlined the importance of early life period in bone development. Although some studies reported no association with bone outcomes, factors affecting prenatal period may have long-term consequences on children’s bone health, as they may act through indirect mechanisms.

In 2010, more than 20 million women and five million men were estimated to have osteoporosis in the European Union and the number of fractures caused by this disease has been estimated as 3.5 million. Moreover, the economic burden of incidents and prior fragility fractures was estimated at € 37 billion in the European Union and a worrying finding is that the costs are expected to increase by 25% by 2025 [89]. This would appear to indicate that osteoporosis should be widely considered as a growing social, economic and health problem. Thus, a decrease in the number of patients with osteoporosis would enable us to save resources in health care and, more importantly, increase the quality of life in the elderly.

This review has underlined the importance of early bone health prophylaxis and actions aimed at ensuring optimal bone development. We identified the need to reinforce communications to young women aimed at emphasizing the importance of the prenatal period for a child’s optimal development.

## Figures and Tables

**Figure 1 nutrients-13-02302-f001:**
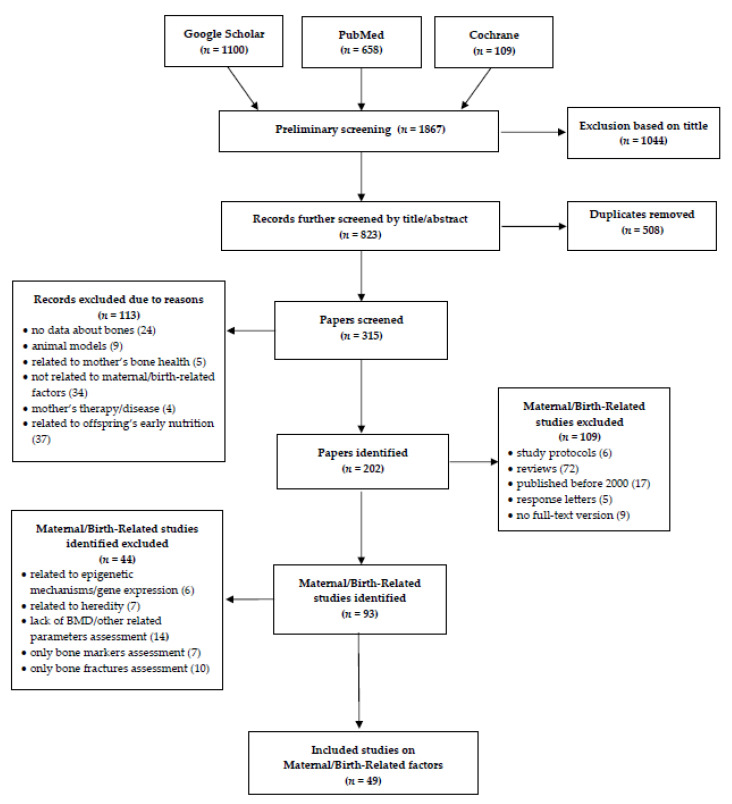
Flow diagram of the records included in the narrative review.

**Table 1 nutrients-13-02302-t001:** Inclusion and exclusion criteria for screening studies.

Inclusion criteria	Studies which aimed to evaluate the association between maternal/birth-related factors and children’s bone outcome Articles available in the English language, in full-text versions Published between January 2000 and August 2020
Exclusion criteria	Lack of data about bone health Animal studies Studies related to maternal bone health/heredity Studies not related to maternal/birth-related factors Studies where mothers were under treatment/had a disease Studies related to offspring’s early nutrition Reviews and meta-analyses Studies published before year 2000Study protocols, conference abstracts, response letters, articles with no full-text available Studies without assessment of BMD (or BMAD/vBMD/aBMD/SOS)

BMD, bone mineral density; aBMD, areal BMD; BMAD, bone mineral apparent density; vBMD, volumetric BMD; SOS, speed of sound.

**Table 2 nutrients-13-02302-t002:** Maternal dietary intake in pregnancy and offspring bone outcomes.

Study Details							Outcome
Country, Year [Reference]	Study Subjects	Study Design	Assessment Method (Age)	Parameters	Dietary Intake Assessment (Gestation)	Components/Nutrients	No Effect/Association (↔) Positive (↑) Negative (↓)
Netherlands, 2013 [19]	F = 1410, 6.14 y * M = 1409, 6.15 y *	Prospective cohort study	DXA (F: 6.14 y * M: 6.15 y *)	TBLH BMD	FFQ (1st trimester)	Macronutrients, Ca, P, Mg (dietary nutrient intake categorized into quintiles; lowest category as the reference category)	TBLH BMD: E from protein (kcal/d), Ca intake (g/d), P intake (g/d) (↑) E from carbohydrate (kcal/d) (↓) E from fat (kcal/d), Mg intake (g/d) (↔)
UK, 2005 [32]	F = 3032 M = 2942 (118 m *)	Prospective cohort study (ALSPAC)	DXA (9 y)	TB and spine BMD	FFQ (32 wk)	Fiber, carbohydrate, starch, sugar, intrinsic/milk sugar, extrinsic non-milk, total fat, saturates, monounsaturates, polyunsaturates, omega-3, protein, Ca, Na, Mg, P, K, total Zn, Fe, retinol, riboflavin, carotene, folate, thiamin, niacin, vitamin B_6_, vitamin C, D, E	Spine BMD: folate (↑)
Australia, 2000 [20]	N = 173 F = 28% M = 72% (8.2 y *)	Longitudinal study	DXA (8.2 y *)	TB, LS and FN BMD	FFQ applied after birth, the reference period was the 3rd trimester	Fat, protein, Mg, K, P density (divided estimated micronutrients intake by the estimated daily E intake or macronutrients intake as % contribution of estimated E intake; split into tertiles)	LS BMD: P density, fat density (↑) TB BMD, FN BMD (↔)
Australia, 2010 [21]	N = 216 F = 30% M = 70% (16.2 y *)	Prospective study	DXA (16.2 y *)	FN, LS and WB BMD	FFQ (3rd trimester)	Protein, fat, carbohydrate, fish, fruit, meat, milk, vegetable, Ca, Mg, P density (divided estimated daily nutrient intake or food intake by estimated total daily energy intake, split into tertiles)	WB BMD (↔) LS BMD: fat density (↓), milk density, Mg density, Ca density (↑) FN BMD: fat density (↓), Mg density (↑)
UK, 2009 [33]	F = 94 M = 104 9 y	Longitudinal study	DXA (9 y)	WB and LS BMD, aBMD	FFQ (15, 32 wk)	High prudent diet score (elevated intakes of fruit, vegetables, wholemeal bread, rice, yoghurt, breakfast cereals, pasta and low intakes of processed foods such as chips, roast potatoes, sugar, white bread, processed meat, crisps, tinned vegetables, soft drinks)	High PDS in early pregnancy: WB BMD (↔) High PDS in late pregnancy: WB BMD (↑)
India, 2006 [24]	F = 326 M = 369 6.2 y **	Prospective cohort study (PMNS)	DXA (6.2 y **)	TB and total spine BMD	FFQ and 24-h recall (18, 28 wk)	Milk and milk products	28 wk: TB BMD (↑), total spine BMD (↔)
UK, 2001 [31]	F = 64 M = 81 (neonates)	Cohort study	DXA (2 d **; 0–13 d)	WB and spine BMD	FFQ (in early and late pregnancy; in the preceding 3 months)	Ca, green vegetables	Early pregnancy Ca intake: spine BMD (↑) green vegetables intake: spine BMD (↓)
Australia, 2017 [30]	F = 177 M = 169 10.9 y	Cohort study (VIP)	DXA (10.9 y)	TBLH and LS BMD	FFQ (28–32 wk; food intake over the previous 12 m)	Protein, Mg, P, Zn, Ca, K	BMD (↔)

aBMD, areal bone mineral density; ALSPAC, Avon Longitudinal Study of Parents and Children; BMD, bone mineral density; DXA, dual energy X-ray absorptiometry; E, energy; FFQ, food frequency questionnaire; FN, femoral neck; LS, lumbar spine; PDS, prudent diet score; PMNS, Pune Maternal Nutrition Study; WB, whole body; TB, total body; TBLH, total body less head; VIP, the Vitamin D in Pregnancy study; F, female; M, male; * mean; ** median; d, days; m, months; wk, week; y, years; if a components/nutrients reported in study details were not mentioned in the column with reported study outcomes, means that it was not significant in the considered analysis.

**Table 3 nutrients-13-02302-t003:** Maternal dietary intervention studies and offspring bone outcomes.

Study Details							Outcome
Country, Year [Reference]	Study Subjects	Study Design	Assessment Method (Age)	Parameters	Nutrients	Intervention (Duration, Dose) Dietary Intake Assessment of Analyzed Nutrients (If Available)	No Effect/Association (↔) Positive (↑) Negative (↓)
UK, 2016 [11]	F = 456 M = 509 (neonates)	Randomized, placebo-controlled trial (MAVIDOS)	DXA (7 d *—placebo; 8 d *—vitamin D)	WB BMD	Vitamin D	14 wk of gestation until delivery 1000 IU/d or placebo	WB BMD (↔)
Iran, 2016 [12]	F = 53 M = 74 (neonates)	Randomized placebo clinical trial	DXA (21.7 d *—vitamin D, 24.5 d *—control)	BMD	Vitamin D	26–28 wk of gestation until delivery 2000 IU/d or placebo Calcium intake (mg/d)—vitamin D vs. control: 1172.58 (467.70), 1190.00 (383.77), respectively (*p* = 0.81) Vitamin D intake (IU/d)—vitamin D vs. control: 2345.16 (240.68), 430.79 (230.80), respectively (*p* < 0.001)	BMD (↔)
Denmark,2020 [13]	F = 256 M = 261 (3 y and 6 y)	Randomized clinical trial (COPSAC_2010_)	DXA (3 y and 6 y)	Total, head, TBLH BMD	Vitamin D	24 wk of gestation until 1 wk after birth 2800 IU/d or 400 IU/d	2800 IU/d vs. 400 IU/d At the age 3 y: Total, head and TBLH BMD (↔) At the age 6 y: Head, total BMD (↑), TBLH BMD (↔) 3 y and 6 y combined: Head BMD (↑), TBLH BMD, total BMD (↔)
Gambia,2017 [14]	F = 231, 9.2 y * M = 216, 9.3 y *	Randomized controlled trial	pQCT	vBMD	Calcium	20 wk of gestation until delivery 1500 mg Ca (Ca carbonate) or placebo Calcium intake (mg/d)—Ca group vs. placebo: 1831 (177), 356 (159), respectively	vBMD (↔)
India, 2017 [15]	Group 1,2—14 m ** Group 3—16 m **	Randomized, placebo-controlled trial	DXA (12–16 m)	WB BMD	Vitamin D + Calcium	14–20 wk of gestation until delivery 60 000 IU cholecalciferol group 1: 4 weekly (+Ca 1g/d) group 2: 8 weekly (+Ca 1g/d) group 3: placebo (only Ca 1g/d + 400 IU cholecalciferol)	Group 3 vs. 1 and 2: WB BMD (↑)
Brazil, 2015 [16]	N = 56 adolescent mother-infant pairs 5 wk	Randomized controlled trial	DXA (5 wk)	TB BMD	Vitamin D + Calcium	26 wk of gestation until delivery cholecalciferol (200 IU/d) + calcium (600 mg/d) or placebo Calcium intake (mg/d)—Ca + vitamin D group vs. placebo: 500 (276), 743 (457), respectively (*p* = 0.02)	TB BMD (↔)
Denmark, 2018 [34]	F = 337 M = 351 6.2 y *	Randomized clinical trial (COPSAC_2010_)	DXA (6.2 y *)	TBLH BMD	n-3 LC PUFAs	24 wk of gestation until 1 wk after birth n-3 LC PUFAs: fish oil (2.4 g; 55% EPA and 37% DHA) control: olive oil (72% n-9 oleic acid and 12% n-6 linoleic acid)	TBLH BMD (↔)

BMD, bone mineral density; vBMD, volumetric bone mineral density; COPSAC2010, Copenhagen Prospective Studies on Asthma in Childhood; DHA, docosahexaenoic acid; DXA, dual energy X-ray absorptiometry; EPA, eicosapentaenoic acid; LC PUFAs, long chain polyunsaturated fatty acids; MAVIDOS, Maternal Vitamin D Osteoporosis Study; TB, total body; TBLH, total body less head; WB, whole body; F, female; M, male; * mean; ** median; d, days; m, months; wk, week; y, years; data about dietary intake presented as mean values (standard deviation).

**Table 4 nutrients-13-02302-t004:** Maternal vitamin D status in pregnancy and offspring bone outcomes.

Study Details						Outcome
Country, Year [Reference]	Study Subjects	Study Design	Assessment Method (Age)	Parameters	Pregnancy Vitamin D Measurement	No Effect/Association (↔) Positive (↑) Negative (↓)
USA, 2019 [17]	F = 118 M = 134 1.7 d *	Longitudinal study (CPEP)	DXA (1.7 d *)	Total, lumbar BMD	25(OH)D in pregnancy (average from 3 measurements: at baseline, 26–29 and 36 wk)	Maternal deficiency vs. non-deficient: Total BMD (↓) Lumbar BMD (↔)
Slovenia, 2019 [35]	F = 34 M = 39 (neonates)	Observational study (My-MILK)	QUS (in the first 48 h after birth)	SOS	25(OH)D in 3rd trimester	SOS (↔)
China, 2010 [42]	F = 119 M = 148 (neonates)	Cross-sectional study	QUS (2.9 d *)	SOS	25(OH)D before delivery	SOS (↑)
Gambia, 2009 [36]	N = 44–52 2–52 wk	Randomized controlled trial (secondary analysis)	DXA (2, 13, 52 wk)	BMD	25(OH)D in 20, 36 wk	BMD (↔)
Finland, 2011 [37]	N = 87 F = 37% M = 57% (14.8 m *)	Prospective cohort study (follow-up)	pQCT (14.8 m *)	left tibia BMD	25(OH)D in 1st trimester	Low D vs. high D: BMD (↔)
Netherlands, 2017 [38]	F = 2663 M = 2631 6.1 y *	Prospective cohort study (the Generation R study)	DXA (6.1 y *)	TBLH BMD	25(OH)D in 20.4 wk **	BMD (↔)
UK, 2006 [18]	F = 94, 8.8 y * M = 104, 8.9 y *	Longitudinal study	DXA (9 y)	WB, LS aBMD	25(OH)D in 34 wk *	WB and LS aBMD (↑)
UK, 2013 [39]	N = 3960 9.9 y *	Prospective cohort study (ALSPAC)	DXA (9.9 y *)	TBLH, spine BMD	25(OH)D at any stage	TBLH, spine BMD (↔)
Australia, 2019 [41]	F = 88 M = 93 10.9 y **	Observational study (VIP)	DXA (10.9 y **)	TBLH and spine BMD, BMAD	25(OH)D at recruitment (<16 wk) and in 28–32 wk	at recruitment: F—BMD (↔) M—BMD (↑) 28-32 wk: BMD (↔)
Australia, 2014 [40]	F = 204, 20.1 y * M = 137, 20.2 y *	Prospective cohort study (the Raine study)	DXA (F: 20.1 y * M: 20.2 y *)	TB BMD	25(OH)D in 18 wk *	BMD (↑)
Norway, 2019 [23]	F = 16 M = 25 26.1 y *	Prospective cohort study (follow-up)	DXA (26.1 y *)	LS, FN, total hip and WB BMD	25(OH)D and 1,25(OH)_2_D in 17, 33, 37 wk	LS, FN, total hip and WB BMD (↔)

aBMD, areal bone mineral density; ALSPAC, Avon Longitudinal Study of Parents and Children; BMAD, bone mineral apparent density; BMD, bone mineral density; DXA, dual energy X-ray absorptiometry; FN, femoral neck; LS, lumbar spine; pQCT, peripheral quantitative computed tomography; SOS, speed of sound; TB, total body; TBLH, total body less head; WB, whole body; VIP, the Vitamin D in Pregnancy study; QUS, quantitative ultrasound; F, female; M, male; * mean; ** median; d, days; m, months; wk, week; y, years.

**Table 5 nutrients-13-02302-t005:** Maternal nutrients status in pregnancy and offspring bone outcomes.

Study Details							Outcome
Country, Year [Reference]	Study Subjects	Study Design	Assessment Method (Age)	Parameters	Nutrients	Nutrient Status Assessment	No Effect/Association (↔) Positive (↑) Negative (↓)
Netherlands, 2013 [19]	F = 1410, 6.14 y * M = 1409, 6.15 y *	Prospective cohort study (the Generation R study)	DXA (F: 6.14 y * M: 6.15 y *)	TBLH BMD	Homocysteine, folate, vitamin B_12_	Concentrations in venous blood (12.9 wk **)	TBLH BMD: Vitamin B_12_ concentration (↑) Homocysteine, folate concentration (↔)
UK, 2013 [43]	F = 342 M = 385 4.1 y **	Prospective cohort study (SWS)	DXA (4.1 y **)	WB/WBMH, LS aBMD, vBMD	LC PUFAs	LC PUFAs (n-3, n-6, EPA, DPA, AA) composition of maternal plasma PC in late pregnancy (34 wk)	Maternal plasma PC concentration: WBMH aBMD: EPA, DPA (↑) LS aBMD: n-3, EPA, DPA, n-6 (↑) WBMH vBMD: EPA (↑) LS vBMD: EPA (↑) Maternal% fatty acids: WBMH aBMD: EPA, DPA (↑), AA (↓) LS aBMD: EPA (↑) WBMH vBMD: EPA (↑) LS vBMD: (↔)
UK, 2016 [22]	F = 241 M = 282 0–2 wk	Prospective birth-cohort study (SWS)	DXA (within 2 wk after birth)	WB BMD	Vitamin A	Blood sample—late pregnancy assessment (34 wk): retinol β-carotene	Retinol: BMD (↔) β-carotene: BMD (↔) β-carotene:retinol ratio BMD (↔)
Norway, 2019 [23]	F = 16 M = 25 26.1 y *	Prospective cohort study (follow-up)	DXA (26.1 y *)	LS, FN, total hip and WB BMD	Vitamin A	Serum samples—all-trans retinol: in pregnancy (17, 33, 37 wk) at birth (cord blood)	17 wk: WB, LS BMD (↑), FN, total hip BMD (↔) 33 wk: BMD at any site (↔) 37 wk: LS, total hip BMD (↑), WB, FN BMD (↔) Retinol in cord blood (↔)
UK, 2006 [18]	F = 94, 8.8 y * M = 104, 8.9 y *	Longitudinal study	DXA (9 y)	WB and LS aBMD	Calcium	Serum sample: umbilical venous blood samples (cord blood)	LS aBMD (↑) WB aBMD (↔)
India, 2006 [24]	F = 326 M = 369 6.2 y **	Prospective cohort study (PMNS)	DXA (6.2 y **)	TB and total spine BMD	Folate	Maternal erythrocyte folate concentrations (18, 28 wk)	28 wk: TB and total spine BMD (↑)

AA, arachidonic acid, aBMD, areal bone mineral density; BMD, bone mineral density; vBMD, volumetric bone density; DPA, docosapentaenoic acid; DXA, dual energy X-ray absorptiometry; EPA, eicosapentaenoic acid; FN, femoral neck; LC PUFAs, long chain polyunsaturated fatty acids; LS, lumbar spine; PC, phosphatidylcholine; PMNS, Pune Maternal Nutrition Study; SWS, Southampton Women’s Survey; TB, total body; TBLH, total body less head; WB, whole body; WBMH, whole body minus head; F, female; M, male; * mean; ** median; wk, week; y, years.

**Table 6 nutrients-13-02302-t006:** Maternal anthropometry and offspring bone outcomes.

Study Details						Outcome
Country, Year [Reference]	Study Subjects	Study Design	Assessment Method (Age)	Parameters	Factors	No Effect/Association (↔) Positive (↑) Negative (↓)
UK, 2010 [52]	F = 398 M = 443 (neonates)	Prospective cohort study (SWS)	DXA (F: 4 d **, M: 5 d **)	WB aBMD	Maternal height	WB aBMD (↔)
UK, 2001 [31]	F = 64 M = 81 (neonates)	Cohort study	DXA (2 d **; 0–13 d)	WB and spine BMD	Maternal height	Spine BMD (↑)
					Maternal BMI (first recorded weight in pregnancy)	WB and spine BMD (↔)
					Maternal triceps skinfold thickness (14 wk)	WB and spine BMD (↑)
Netherlands, 2011 [53]	F = 107, 6.3 m * M = 145, 6.4 m *	Prospective cohort study (the Generation R study)	DXA (6 m)	TB and LS BMD, LS BMAD	Maternal height	LS BMAD (↓) TB and LS BMD (↔)
					Maternal BMI (in pregnancy)	LS BMD (↑) TB BMD, LS BMAD (↔)
					Maternal pregnancy weight gain	TB BMD (↑) LS BMD, BMAD (↔)
India, 2006 [24]	F = 326 M = 369 6.2 y **	Prospective cohort study (PMNS)	DXA (6.2 y **)	TB and total spine BMD	Maternal height	TB and total spine BMD (↔)
Sweden, 2012 [54]	M = 1009 (18.9 y *)	Cohort study (GOOD)	DXA (18.9 y *)	LS aBMD (i.a.)	Maternal height	LS aBMD (↔)
Portugal, 2019 [55]	F = 1014 M = 1153 7 y	Birth-cohort study	DXA (7 y)	WBLH aBMD	Maternal BMI (self-reported weight at the beginning of pregnancy or on the first prenatal medical visit) GWG (the difference between the mother’s self-reported pre-delivery weight and her early pregnancy weight)	In under/normal weight mothers GWG (↑) WBLH aBMD In overweight/obese women GWG (↔) WBLH aBMD
China, 2013 [56]	F = 5306 M = 6592 9.3 m *	Cross-sectional study	DXA (6.7 m **)	LS BMD	Maternal pregnancy weight gain	LS BMD (↑)

aBMD, areal bone mineral density; BMAD, bone mineral apparent density; BMD, bone mineral density; BMI, body mass index; DXA, dual energy X-ray absorptiometry; GOOD, Gothenburg Osteoporosis and Obesity Determinants study; GWG, gestational weight gain; LS, lumbar spine; PMNS, Pune Maternal Nutrition Study; SWS, Southampton Women’s Survey; TB, total body; WBLH, whole body less head; WB, whole body; F, female; M, male; * mean; ** median; d, days; m, months; wk, week; y, years.

**Table 7 nutrients-13-02302-t007:** Maternal prepregnancy weight and offspring bone outcomes.

Study Details						Outcome
Country, Year [Reference]	Study Subjects	Study Design	Assessment Method (Age)	Parameters	Weight Assessment Details	No Effect/Association (↔) Positive (↑) Negative (↓)
USA, 2015 [57]	F = 167 M = 158 0–6 y	Longitudinal study	DXA (at 0.25, 0.5, 0.75, 1, 2, 3, 4, 5, and 6 y)	WB BMD	Prepregnancy weight self-reported	WB BMD (↔)
UK, 2010 [59]	F = 3591 M = 3530 118 m *	Prospective cohort study (ALSPAC)	DXA (9.9 y *)	TBLH and spine BMD	Reported by the mother in a questionnaire administered during pregnancy	TBLH and spine BMD (↔)
Japan, 2020 [58]	F = 375 M = 392 10 y	Retrospective cohort study (JKB)	DXA 10 y	TBLH aBMD	Weight at the beginning of pregnancy when not much weight had been gained	TBLH aBMD (↔)
Sweden, 2012 [54]	M = 1009 (18.9 y *)	Cohort study (GOOD)	DXA (18.9 y *)	LS aBMD (i.a.)	Maternal weight before pregnancy	LS aBMD (↑)

aBMD, areal bone mineral density; ALSPAC, Avon Longitudinal Study of Parents and Children; BMD, bone mineral density; DXA, dual energy X-ray absorptiometry; GOOD, Gothenburg Osteoporosis and Obesity Determinants study; JKB, Japan Kids Body-Composition study; LS, lumbar spine; TBLH, total body less head; WB, whole body; F, female; M, male; * mean; m, months; y, years.

**Table 8 nutrients-13-02302-t008:** Maternal demographic and socioeconomic factors and offspring bone outcomes.

Study Details						Outcome
Country, Year [Reference]	Study Subjects	Study Design	Assessment Method (Age)	Parameters	Factors	No Effect/Association (↔) Positive (↑) Negative (↓)
Sweden, 2012 [54]	M = 1009 (18.9 y *)	Cohort study (GOOD)	DXA, pQCT (18.9 y *)	DXA: TB, FN, LS and radius non-dominant aBMD pQCT: radius cortical and trabecular vBMD	Maternal age	LS aBMD (↓) TB, FN, radius non-dominant aBMD (↔) vBMD at any site (↔)
					Parity	LS aBMD (↔)
USA, 2009 [60]	M = 24 6.9–7.4 y *	Cross-sectional study	DXA, pQCT (6.9–7.4 y *)	DXA: hip, spine, FN aBMD pQCT: distal tibia cortical and trabecular vBMD	Maternal education	(↔)
Lebanon, 2008 [61]	F = 156 M = 170 13.1 y *	Observational study	DXA (~13.1 y *)	TB, LS, FN, total hip BMD	Maternal SES	F: BMD at any site (↑) M: TB, FN, total hip BMD (↑) LS BMD (↔)
UK, 2010 [52]	F = 398 M = 443 (neonates)	Prospective cohort study (SWS)	DXA (F: 4 d ** M: 5 d **)	WB aBMD	Parity	WB aBMD (↔)
USA, 2003 [62]	F = 50 M = 45 (neonates; born < 37 wk)	Prospective study	QUS (within the first 10 d of life)	SOS	Parity	SOS (↔)
India, 2006 [24]	F = 326 M = 369 6.2 y **	Prospective cohort study (PMNS)	DXA (6.2 y **)	TB and total spine BMD	Parity	TB and total spine BMD (↔)

aBMD, areal bone mineral density; BMD, bone mineral density; vBMD, volumetric bone density; DXA, dual energy X-ray absorptiometry; GOOD, Gothenburg Osteoporosis and Obesity Determinants study; FN, femoral neck; LS, lumbar spine; pQCT, peripheral quantitative computed tomography; PMNS, Pune Maternal Nutrition Study; SES, socioeconomic status; SOS, speed of sound; SWS, Southampton Women’s Survey; TB, total body; WB, whole body; QUS, quantitative ultrasound; F, female; M, male; * mean; ** median; d, days; wk, week; y, years.

**Table 9 nutrients-13-02302-t009:** Maternal smoking in pregnancy and offspring bone outcomes.

Study Details					Outcome
Country, Year [Reference]	Study Subjects	Study Design	Assessment Method (Age)	Parameters	No Effect/Association (↔) Positive (↑) Negative (↓)
UK, 2001 [31]	F = 64 M = 81 (neonates)	Cohort study	DXA (2 d **; 0–13 d)	WB and spine BMD	WB BMD (↓)
UK, 2010 [52]	F = 398 M = 443 (neonates)	Prospective cohort study (SWS)	DXA (F: 4 d ** M: 5 d **)	WB aBMD	WB aBMD (↔)
Netherlands, 2015 [66]	F = 2520 M = 2466 6–6.1 y **	Prospective cohort study (the Generation R study)	DXA (6–6.1 y **)	BMD	BMD (↔)
UK, 2011 [67]	F = 3589 M = 3532 9.9 y *	Prospective birth-cohort study (ALSPAC)	DXA (118.4 m **)	TBLH and spine BMD	TBLH and spine BMD (↔)
Australia, 2013 [64]	F = 150 M = 265 16.3 y *	Longitudinal study	DXA (16.3 y *)	Spine, hip, radius, TB BMD	BMD at any site (↔)
Brazil, 2014 [65]	F = 1563 M = 1512 (follow up at 18 y)	Birth-cohort study	DXA (18 y)	BMD	BMD (↔)
Sweden, 2012 [54]	M = 1009 (18.9 y *)	Cohort study (GOOD)	DXA (18.9 y *)	LS aBMD (i.a.)	LS aBMD (↔)
Australia, 2020 [63]	F = 74 M = 122 25.3 y *–25.6 y *	Birth-cohort study	DXA, HRpQCT (term—25.6 y *, preterm—25.3 y *)	aBMD, vBMD, bone microarchitecture	Preterm (↔) Term: Tb.N (↓) Inner TZ porosity (↑)

aBMD, areal bone mineral density; ALSPAC, Avon Longitudinal Study of Parents and Children; BMD, bone mineral density; vBMD, volumetric bone density; DXA, dual energy X-ray absorptiometry; GOOD, Gothenburg Osteoporosis and Obesity Determinants study; HRpQCT, high-resolution peripheral quantitative computed tomography; SWS, Southampton Women’s Survey; TB, total body; TBLH, total body less head; Tb.N, trabecular number; TZ, transitional zone; WB, whole body; F, female; M, male; * mean; ** median; d, days; m, months; y, years.

**Table 10 nutrients-13-02302-t010:** Birth anthropometry and offspring bone outcomes.

Study Details						Outcome
Country, Year [Reference]	Factors	Study Subjects	Study Design	Assessment Method (Age)	Parameters	No Effect/Association (↔) Positive (↑) Negative (↓)
USA, 2003 [62]	Birth anthropometry	F = 50 M = 45 (neonates; born < 37 wk)	Prospective study	QUS (within the first 10 d of life)	SOS	SOS: Birth weight, length (↑)
UK, 2001 [31]	Birth anthropometry	F = 64 M = 81 (neonates)	Cohort study	DXA (2 d **; 0–13 d)	WB and spine BMD	Birth weight: WB, spine BMD (↑) Ponderal index: WB BMD (↑), spine BMD (↔)
China, 2013 [56]	Birth anthropometry	F = 5306 M = 6592 9.3 m *	Cross-sectional study	DXA (6.7 m **)	LS BMD	Birth weight: LS BMD (↑) Birth length: LS BMD (↔)
Australia, 2000 [70]	Birth anthropometry	F = 115, 8.26 y * M = 215, 8.17 y *	Longitudinal study	DXA (F: 8.26 y * M: 8.17 y *)	LS and FN BMD, BMAD	Birth weight: BMD and BMAD at any site (↔) Birth length: LS BMD (↑) FN BMD, BMAD at any site (↔)
India, 2006 [24]	Birth length	F = 326 M = 369 6.2 y **	Prospective cohort study (PMNS)	DXA (6.2 y **)	TB and total spine BMD	TB and total spine BMD (↔)
Turkey, 2006 [71]	Birth weight	F = 50 M = 50 (neonates)	Cross-sectional study	DXA (within first 24 h after birth)	WB BMD	SGA < AGA < LGA
China, 2010 [42]	Birth weight	F = 119 M = 148 (neonates)	Cross-sectional study	QUS (2.9 d *)	SOS	<1500 g vs. ≥2500 g SOS (↓)
Netherlands, 2011 [53]	Birth weight	F = 107, 6.3 m * M = 145, 6.4 m *	Prospective cohort study (the Generation R study)	DXA (6 m)	TB and LS BMD, LS BMAD	TB BMD (↑) LS BMD, LS BMAD (↔)
Netherlands, 2014 [10]	Birth weight	F = 2732 M = 2718 (6 y **)	Prospective cohort study (the Generation T study)	DXA (6 y **)	WB/WBLH BMD	SGA, AGA, LGA: BMD (↔) ≥ 2500–3000 g vs. ≥ 3000–3500 g BMD (↓)
South Africa, 2006 [72]	Birth weight	F = 54 M = 55 8.1 y *	Cohort study (follow-up study)	QUS (8.1 y *)	SOS	SOS (↔)
Australia, 2013 [64]	Birth weight	F = 150 M = 265 16.3 y *	Longitudinal study	DXA (16.3 y *)	Spine, hip, radius, TB BMD	BMD at any site (↔)
Brazil, 2014 [65]	Birth weight	F = 1563 M = 1512 (follow up at 18 y)	Birth-cohort study (The 1993 Pelotas Birth Cohort)	DXA (18 y)	BMD	BMD (↑)
Sweden, 2012 [54]	Birth weight	M = 1009 (18.9 y *)	Cohort study	DXA (18.9 y *)	LS aBMD (i.a.)	LS aBMD (↔)
Australia, 2020 [63]	Birth weight	F = 74 M = 122 25.3 y *–25.6 y *	Birth-cohort study	DXA, HRpQCT (term–25.6 y *, preterm–25.3 y *)	aBMD, vBMD, bone microarchitecture	(↔)
Norway, 2019 [23]	Birth weight	F = 16 M = 25 26.1 y *	Prospective cohort study (follow-up)	DXA (26.1 y *)	LS, FN, total hip and WB BMD	BMD at any site (↔)

AGA/LGA/SGA, appropriate/large/small for gestational age; aBMD, areal bone mineral density; BMAD, bone mineral apparent density; BMD, bone mineral density; vBMD, volumetric bone density; DXA, dual energy X-ray absorptiometry; FN, femoral neck; HRpQCT, high-resolution peripheral quantitative computed tomography; LS, lumbar spine; PMNS, Pune Maternal Nutrition Study; SOS, speed of sound; TB, total body; WB, whole body; WBLH, whole body less head; QUS, quantitative ultrasound; F, female; M, male; * mean; ** median; d, days; m, months; wk, week; y, years.

## Data Availability

No new data were created or analyzed in this study. Data sharing is not applicable to this article.

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
