# Peer review of "Maternal Diet, Nutritional Status, and Birth-Related Factors Influencing Offspring’s Bone Mineral Density: A Narrative Review of Observational, Cohort, and Randomized Controlled Trials"

_nutrients, 2021, doi:10.3390/nu13072302_

Round 1
Reviewer 1 Report
This work is really interesting and represents a meta-analysis of the literature on this topic and I believe it is delegante for this field of research and for the potential translability of the collected data into the clinical practice. Indeed, a good quality of bone in the childhood and in adolescence represents a valuable passport for the future. The manuscript is valuable for pubblication.
Author Response
Dear Reviewer,
Thank you for all your work on our manuscript “Maternal Diet, Nutritional Status, and Birth-Related Factors Influencing Offspring’s Bone Mineral Density: A Narrative Review of Observational, Cohort, and Randomized Controlled Trials” (previously: “Maternal Diet, Nutritional Status and Birth-Related Factors Influencing Offspring’s Bone Mineral Density: A Narrative Review”). Your comments and suggestions were very useful and helped to improve the paper considerably. All your suggestions have been taken into account in the recent revision of the manuscript. You can find answers to your specific comments below.
Yours Sincerely,
Jadwiga Hamułka

Reviewer 2 Report
In this manuscript the authors wanted to elaborate on maternal diet, nutritional status and birth related factors on the bone mineral density in the off springs. They did a comprehensive literature search included key words like maternal, pregnancy, vitamin D supplementation, diet, bone mass and other co-morbitities including BMD and obviously they selected the term offspring. Although the study is important given that we know that these factors may affect the peak bone density and ultimately affect the BMD later in life. Unfortunately the data presented is very condensed and at the end I feel the audience will not learn any specific concept from the study. Due to this ambiguous nature I feel that the paper should drastically summarized and more emphasis should be made on factors which are positively related to the bone density. Secondly, as they alluded themselves, areal BMD is not a reliable tool to access BMD in growing children. Thus, any positive conclusion with relationship to nutritional and environmental factors will remain unanswered. Third, again as they alluded the inconsistency of the result maybe in part due to the use of different methods accessing the food intake and other related factors. Fourth, they should mention what distinguished this work from the previous study related to the same subject. In conclusion, the work has significant methodological deficiencies and I feel that the hypothesis is not really tested.
Author Response

(The authors gave the same response as above.)

Reviewer 3 Report
This narrative review addresses important and unresolved issues regarding the early life factors (in this case focused on maternal factors in pregnancy/at birth) that influence skeletal development and peak bone mass in children. The project was designed as a systematic review but reported as a narrative review and thus does not really contribute to the resolution of outstanding questions. It is appreciated that the review and documentation of the papers cited is a tremendous undertaking, but the lack of rigour in the analysis of the findings limits its value.
Title: Should indicate that studies are a mix of research designs – observational, cohort and RCT
Methods
L69 onward - The PRISMA approach to the literature search as noted in Figure 1 seems appropriate but the project was not completed as a systematic review and meta analysis. Not only was the search not exhaustive (as noted in line 442) but risk of bias assessment was not conducted nor was meta analysis of the findings across papers within each sub-topic. The exact process used needs to be clarified in the methods and changed in the figure.
Terminology – Why was BMD used as the primary outcome rather than BMC or BMC z-scores? As both bone mineral content and bone area increase with age, the BMD measure is not as sensitive as BMC for age as a measure of bone mineral accretion.
Results
L96 – It is not usual to combine results (Findings) and discussion sections.
L104-126 – Data on the association of individual nutrients in maternal diet during pregnancy and infant bone outcomes would be more meaningful if the absolute intake of nutrients were reported. Some studies may be reporting dietary inadequacies while others dietary adequacy or even excess. For the dietary intake of individual nutrients, there is no mention of the amount of nutrients studied. At the very least, the studies should be separated into those in which nutrient intake met recommended intakes (such as the Dietary Reference Intake recommendations) and those in which intakes were inadequate in relation to recommendations. For example, for vitamin D, abnormalities in bone mass of the offspring are usually associated with sub-optimal maternal vitamin D intake or status but not when intakes are adequate.
L196-7 – Is this supposed to be a sentence or a heading. If the latter then it should be in italics or otherwise distinguished from the text.
L199-202 – Sentence requires better clarity in meaning.
L422-432 – This paragraph does not adequately summarize the findings in terms of implications.
Editing comments
- Many grammatical errors throughout the paper including improper tense, plurality, adverbs, spelling etc
Review
This narrative review addresses important and unresolved issues regarding the early life factors (in this case focused on maternal factors in pregnancy/at birth) that influence skeletal development and peak bone mass in children. The project was designed as a systematic review but reported as a narrative review and thus does not really contribute to the resolution of outstanding questions. It is appreciated that the review and documentation of the papers cited is a tremendous undertaking, but the lack of rigour in the analysis of the findings limits its value.
Title: Should indicate that studies are a mix of research designs – observational, cohort and RCT
Methods
L69 onward - The PRISMA approach to the literature search as noted in Figure 1 seems appropriate but the project was not completed as a systematic review and meta analysis. Not only was the search not exhaustive (as noted in line 442) but risk of bias assessment was not conducted nor was meta analysis of the findings across papers within each sub-topic. The exact process used needs to be clarified in the methods and changed in the figure.
Terminology – Why was BMD used as the primary outcome rather than BMC or BMC z-scores? As both bone mineral content and bone area increase with age, the BMD measure is not as sensitive as BMC for age as a measure of bone mineral accretion.
Results
L96 – It is not usual to combine results (Findings) and discussion sections.
L104-126 – Data on the association of individual nutrients in maternal diet during pregnancy and infant bone outcomes would be more meaningful if the absolute intake of nutrients were reported. Some studies may be reporting dietary inadequacies while others dietary adequacy or even excess. For the dietary intake of individual nutrients, there is no mention of the amount of nutrients studied. At the very least, the studies should be separated into those in which nutrient intake met recommended intakes (such as the Dietary Reference Intake recommendations) and those in which intakes were inadequate in relation to recommendations. For example, for vitamin D, abnormalities in bone mass of the offspring are usually associated with sub-optimal maternal vitamin D intake or status but not when intakes are adequate.
L196-7 – Is this supposed to be a sentence or a heading. If the latter then it should be in italics or otherwise distinguished from the text.
L199-202 – Sentence requires better clarity in meaning.
L422-432 – This paragraph does not adequately summarize the findings in terms of implications.
Editing comments
- Many grammatical errors throughout the paper including improper tense, plurality, adverbs, spelling etc
-
- Many grammatical errors throughout the paper including improper tense, plurality, adverbs, spelling etc
Author Response

(The authors gave the same response as above.)

Reviewer 4 Report
The authors conducted a review that tryed to reveal if the maternal and birth-related factors may influence children bone mineral density.
The review was carefully prepared and the presented studies are interesting and focused on the main topic. The presentation of the results are novel and reliable and highlighted the inconsistency in the foregoing studies.
Furthermore this review has underlined the importance of early bone health prophylaxis and actions aimed to ensure optimal bone development. But as in this type of work some mistakes have been made: not all associeated factors are shown that may contribute to BMD. Moreover, some associations for example mechanisms of actions of vit. D and it's metabolites (which type of metabolites vere messured in the reserches and type of activity actions for exemple) were shown and discussed. However, despite the above-metioned accusations I rate the work higly.
Author Response

(The authors gave the same response as above.)

Round 2
Reviewer 3 Report
NUTRIENTS re-review – Narr review of factors affecting BMD in kids
The authors have addressed many of the issues that I raised in my initial review, although some of questions appeared to unanswerable given the details of information in the papers reviewed, such as regarding the details of the dietary intakes.
Some issues remain outstanding:
- L137-43 and Table 2 – Better clarity is required as to what is meant by quantitative nutrient intake and these require better definition in the table. For example, the following paragraph from L137-143 would provide better clarity if written as follows if I am understanding the units of measure alluding to in the paper (my changes in bold):
Maternal carbohydrate intake as % total energy intake was negatively associated with offspring BMD [19]. In one study % energy as protein intake was positively related to BMD in offspring [19], however in two other studies, no association between protein intake was found [20,30]. Results regarding fat expressed as % total energy intake were ambiguous: one study found a negative association between fat density and offspring lumbar spine (LS) and femoral neck (FN) BMD (with no changes in whole body (WB) BMD) [21], one study found a positive association [20], and one study reported no association [19]
- Table 2 – need to better define what is mean by fat density (I assume fat as % total energy intake), E from protein and milk density.
- L148-9 – A similar issue applies to this section which I suggest is written as:
In some studies, positive associations between offspring BMD (in at least one site) was associated with higher maternal intakes of magnesium [21], 148 phosphorus [19,20], calcium [19,21,31], or folate [32] intake in pregnancy.
- L167-180 – better define the intake ranges of calcium, phosphorus and vitamin D that were studied as noted in Table 3.
- L535-49 – The text does not provide key highlights of the findings of the paper. Thus, it is not a very effective final statement nor does it provide an meaningful take home message.
Editing
While the authors indicated in their rebuttal that the paper has been edited by a professional English speaking person, errors in grammar and awkward English composition still exist.
Author Response
Dear Reviewer,
We would like to thank you again for all your work on our manuscript “Maternal Diet, Nutritional Status and Birth-Related Factors Influencing Offspring’s Bone Mineral Density: A Narrative Review”. Your further comments and suggestions were very useful and helped to improve the paper considerably. All your suggestions have been taken into account in the recent revision of the manuscript. You can find answers to your specific comments below.
Sincerely,
Jadwiga Hamulka
